# Agent-based modeling reveals how bats navigate dense group emergences

Omer Mazar[1]*, Yossi Yovel[1,2]

[1]Tel Aviv University, Sagol School of Neuroscience, Tel Aviv, Israel; [2]Tel Aviv University, Faculty of Life Sciences, School of Zoology, Tel Aviv, Israel

## eLife Assessment

This **important** model-based study seeks to mimic bat echolocation behavior and flight under conditions of high interference, such as when large numbers of bats leave their roost together. The simulations **convincingly** suggest that the problem of acoustic jamming in these situations may be less severe than previously thought. This finding will be of broad interest to scientists working in the fields of bat biology and collective behaviour.

*For correspondence:
omer_mazar@yahoo.com

Competing interest: The authors declare that no competing interests exist.

**Abstract** Bats face a complex navigation challenge when emerging from densely populated roosts, where vast numbers take off at once in dark, confined spaces. Each bat must avoid collisions with walls and conspecifics while locating the exit, all amidst overlapping acoustic signals. This crowded environment creates the risk of acoustic jamming, in which the calls of neighboring bats interfere with echo detection, potentially obscuring vital information. Despite these challenges, bats navigate these conditions with remarkable success. Although bats have access to multiple sensory cues, here, we focused on whether echolocation alone could provide sufficient information for orientation under such high-interference conditions. To explore whether and how they manage this challenge, we developed a sensorimotor model that mimics the bats' echolocation behavior under high-density conditions. Our model suggests that the problem of acoustic jamming may be less severe than previously assumed. Frequent calls with short inter-pulse intervals (IPI) increase the sensory input flow, allowing integration of echoic information across multiple calls. When combined with simple movement-guidance strategies—such as following walls and avoiding nearby obstacles—this accumulated information enables effective navigation in dense acoustic environments. Together, these findings demonstrate a plausible mechanism by which bats may overcome acoustic interference and underscore the role of signal redundancy in supporting robust echolocation-based navigation. Beyond advancing our understanding of bat behavior, they also offer valuable insights for swarm robotics and collective movement in complex environments.

## Introduction

In many bat species, individuals dwell together in caves (or similar roosts), forming large colonies with tens to several millions of individuals (*Glover and Altringham, 2008*; *Hristov et al., 2010*). Each evening, at approximately the same time, the bats take off from their roost, navigating through its passages toward the exit. The high density of bats flying simultaneously in great proximity poses many challenges for orientation in such a crowded and noisy environment. Flying while avoiding collisions, often in a pitch-black cave, demands the continuous detection and localization of both obstacles and nearby bats (*Vanderelst et al., 2015*; *Gillam et al., 2010*). Employing echolocation, bats emit strong ultrasonic signals and interpret the reflected echoes to perceive their surroundings (*Ulanovsky and Moss, 2008*). The reception of neighbors' loud calls, which share similar acoustic features with their

**eLife digest** Bats rely on echolocation to navigate and hunt in complete darkness. They emit high-frequency calls and use the returning echoes to detect the distance, shape and location of objects around them. However, when many bats are flying in the same area and calling at the same time – such as during group takeoffs from a crowded cave – the overlapping calls can interfere with each other, creating a phenomenon known as acoustic jamming. This interference may block important echoes, making it harder to detect obstacles or find a way out.

Scientists have long questioned how bats manage to avoid collisions and successfully navigate in such noisy, crowded conditions. If echolocation is still effective under such conditions, what strategies might these animals use to overcome interference? To investigate this, Mazar and Yovel developed a computer model that simulates how individual bats might behave and perceive their environment when flying in large, acoustically noisy groups.

The simulations showed that acoustic jamming is less problematic than previously assumed. Frequent call emission allows bats to collect redundant sensory information, and integrating these signals over short periods helps them form a reliable picture of their surroundings. In addition, simple behavioral strategies, such as following walls and avoiding collisions with other bats, allow them to navigate safely. The researchers also found that an agent-based model using signal redundancy, short-term memory, and local movement rules can robustly replicate the dynamics seen in real bats. These findings suggest that bats have evolved behavioral and sensory adaptations that make echolocation effective even in dense, noisy conditions.

These findings can benefit researchers studying bat behavior, sensory processing and neural adaptation to complex environments. They may also inform the design of drone swarms or autonomous agents that need to operate in crowded, noisy conditions. Importantly, this study shows how simulations, particularly agent-based models, are powerful tools for exploring complex group behavior and sensory challenges. To apply these insights practically, further empirical validation and real-world implementation in robotics or neuroscience are needed.

own calls, can potentially hinder the bats' ability to detect the faint echoes reflected off the walls and the surrounding bats (*Ulanovsky and Moss, 2008*; *Lin et al., 2016*). We examined whether bats could rely solely on echolocation to exit the roost even during such a chaotic 'rush hour'.

The question of how bats cope with acoustic interference—i.e., the masking of potential echoes by conspecific signals—has been extensively researched using playback experiments, field observations, on-body tags, and computational simulations (*Mazar and Yovel, 2020*; *Takahashi et al., 2014*; *Ulanovsky et al., 2004*; *Gillam and Montero, 2016*; *Bates et al., 2008*; *Luo and Moss, 2017*; *Corcoran and Conner, 2014*; *Cvikel et al., 2015a*; *Cvikel et al., 2015b*; *Götze et al., 2016*; *Fawcett et al., 2015*). However, much of this research has focused on foraging bats in small groups (*Ulanovsky and Moss, 2008*; *Lin et al., 2016*; *Ulanovsky et al., 2004*; *Götze et al., 2016*; *Gillam et al., 2007*; *Giuggioli et al., 2015*; *Obrist, 1995*). The challenges bats encounter during roost exits (e.g. cave exits) differ markedly from those encountered during group foraging. Bat density during roost exits is significantly higher, and bats need to detect and follow static walls or obstacles, which produce loud echoes, rather than small, sporadic prey items that generate faint echoes (*Krivoruchko et al., 2024*). Their flight during exits is also more directional and involves avoiding collisions with conspecifics, in contrast to the erratic hunting maneuvers typically observed while foraging. Echolocation studies during dense collective movement are scarce (*Gillam et al., 2010*; *Lin et al., 2016*; *Hristov et al., 2008*; *Lin and Abaid, 2015*; *Beleyur and Goerlitz, 2019*; *Goldshtein et al., 2025*), likely due to the complexities in recording separate echolocation calls and tracking individual flights within the swarm.

While collective movement has been extensively studied in various species, such as insect swarming, fish schooling, and bird murmuration (*Pearce et al., 2014*; *Bastien and Romanczuk, 2020*; *Davidson et al., 2021*; *Aidan et al., 2024*; *Pitcher et al., 1976*; *Partridge, 1982*; *Strandburg-Peshkin et al., 2013*), as well as in swarm robotics, where agents perform tasks, such as coordinated navigation and maze-solving (*Youssefi and Rouhani, 2021*; *Cheraghi et al., 2021*; *Dias et al., 2021*), most studies have focused on movement algorithms that assume full detection of neighbors (*Couzin et al., 2002*; *Attanasi et al., 2014*; *Gautrais et al., 2012*; *Nagy et al., 2010*; *Parrish and Edelstein-Keshet, 1979*;

*Sumpter et al., 2008*; *Bialek et al., 2012*; *Couzin et al., 2005*). Some models have incorporated limited interaction rules where individuals respond to only one or a few neighbors due to sensory constraints (*Bode et al., 2011*; *Jhawar et al., 2020*). However, fewer studies have explicitly examined how sensory interference, occlusion, and noise influence decision-making and affect collective movement (*Rosenthal et al., 2015*).

The present study addresses these gaps by introducing an agent-based sensorimotor model based on the well-documented echolocation capabilities of bats, simulating multiple bats pathfinding their way out of a cave-like structure. We modeled the echolocation behavior of two insectivorous bat species: Pipistrellus kuhlii (PK), which roosts in abandoned buildings and frequently navigates through conspecific-dense, cluttered corridors, and the cave-dwelling Rhinopoma microphyllum (RM), which emerges from its roosts with thousands of individuals simultaneously. These two species differ in their echolocation signals - PK echolocation signals are characterized by a wider bandwidth and a higher terminal frequency than RM calls. We quantified the performance of an individual bat flying among conspecifics, demonstrating that even a relatively simple sensorimotor algorithm can facilitate successful orientation in such complex environments. The modeling approach enabled us to explore how various biological and ecological factors influence successful navigation under such challenging conditions.

## Results

Our model was designed with conservative assumptions regarding bats' sensing, movement, and sensorimotor integration, aiming to underestimate their capabilities and thereby establish a lower bound on their actual performance. Real bats thus likely outperform the model's predictions. In our 2D simulations (*Mazar and Yovel, 2020*), each bat emits sound signals and receives echoes reflected from the roost walls and other bats, while also encountering masking signals caused by calls from conspecifics. These masking signals can interfere with and completely eliminate echo detection (which we refer to as jamming) or cause echo localization errors. After estimating the distance and direction of each detected reflector, the bat adjusts its echolocation parameters and maneuvers to find the exit while simultaneously avoiding collisions. The bats dynamically adjust their echolocation parameters— including call rate, duration, and frequency range—based on the estimated distance to obstacles, following the well-documented transition between search, approach, and buzz phases observed in echolocating bats (see *Mazar and Yovel, 2020* and Methods). Their reception was modeled using a biologically inspired filter-bank receiver comprising 80 gammatone channels (*Mazar and Yovel, 2020*; *Sanderson et al., 2003*; *Neretti et al., 2003*). Each bat adjusted its flight following a simple path-finding algorithm based solely on the estimated locations of the detected reflectors (see Methods, *Figure 1—figure supplement 1*, and *Figure 1—video 1* for additional details). The bats had to exit a roost designed as a corridor (14.5 m long × 2.5 m wide), with a right-angle turn located 5.5 m before the exit (*Figure 1A*). Additionally, an obstacle (1.25 m wide) was situated 2.25 m in front of the opening. The simulated bats initiated their flight from the far end of the corridor, within a randomly selected 1.5 × 2 m² area, taking off in the general direction of the exit (±30 degrees), without prior knowledge of the roost's structure.

The sensory model accounted for six types of acoustic signals: (1) the bat's own calls, (2) echoes from conspecifics, (3) echoes from walls in response to the bat's own calls (i.e. desired wall echoes), (4) echoes from conspecific calls reflected off other bats, (5) echoes from conspecific calls reflected off walls, and (6) the conspecific calls themselves. In the baseline model, bats were assumed to reliably distinguish between all these signal types. In contrast, the confusion model described below specifically tested the impact of failing to distinguish between desired wall echoes (3) and wall echoes generated by conspecific calls (5), while preserving the bat's ability to identify all other signal types. In brief, the bat responded to echoes as follows (see Methods and *Figure 1—figure supplement 1* for details): If an obstacle or a conspecific was detected in front of the bat and was too close, the bat would maneuver to avoid a collision. Otherwise, for exit-seeking, the bat would follow the contour of the walls by steering toward the farthest detected obstacle ahead. If a gap greater than 0.5 m was identified between adjacent reflectors, the bat directed its trajectory toward the center of the gap.

The ability of the bats to exit the roost within 15 s was evaluated for different group sizes, from a single bat and up to 100 individuals. For simplicity, we will refer to the initial density at the cave's far end as the number of bats per 3 m² (i.e. when we refer to a group of 100 bats, the density is 100

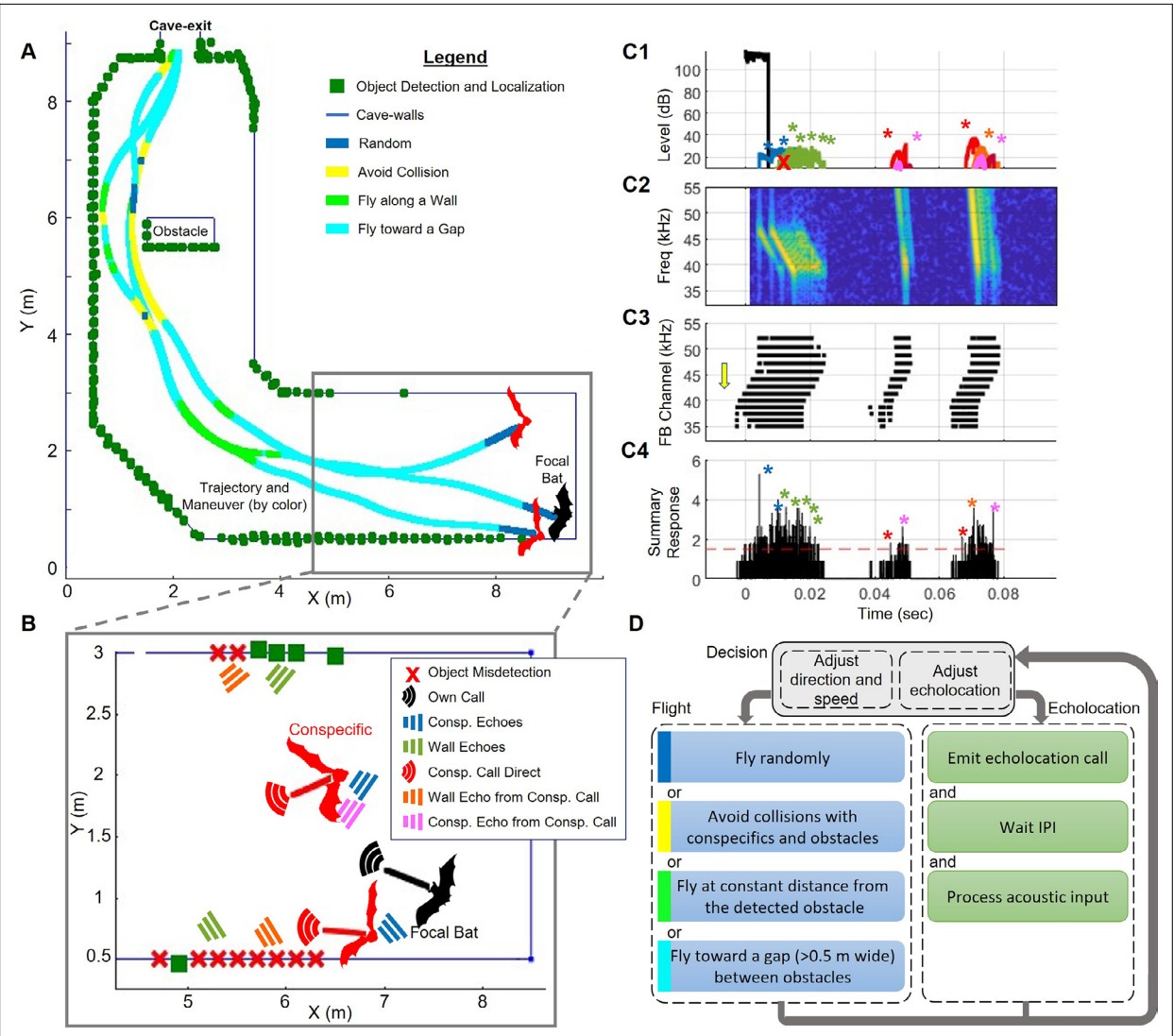

**Figure 1.** The sensorimotor model. (**A**) Top view of the cave with three bats' trajectories. The focal bat is shown in black. All bats' flight trajectories are displayed while the bats' moment-to-moment decisions are represented by the colored lines: blue - random flight, yellow - collision avoidance, light green - wall-following, turquoise - movement toward a wall gap (see panel D for details). Green squares depict reflectors detected by the focal bat along its route. (**B**) A zoomed-in view of the marked rectangular area in panel A, where the focal bat (black) emitted one echolocation call (black) and received echoes from the cave walls (green) and from two other bats (blue). It also received conspecifics' calls (red) and their reflection from the cave walls (orange), as well as the reflections from other bats (pink). Green squares indicate points that were detected by the focal bat from this call, and red x's indicate missed points due to acoustic masking (i.e. jammed reflectors). The locations of the detected reflectors (green squares) are marked according to their localization by the bat (with simulated errors). The lines near the bats depict their flight direction. (**C**) The acoustic scene received by the focal bat is as depicted in **B**, including the emitted call and all received signals (colors as in panel **B**). (**C1**) The time-domain plot displays the envelope of signals, encompassing the emitted call and the received signals: the desired echoes from the walls and conspecifics; the calls of other bats; the echoes returning from conspecific calls and reflected off the walls and off other bats. Notably, in this example, some of the desired wall-echoes are jammed by stronger self-echoes reflected from nearby conspecifics. (**C2**) The spectrogram of all the received signals presented in **C1**; for clarity, the emitted call is not depicted. (**C3**) The responses of the active channels of the cochlear filter bank (FB channel) after de-chirping. Each channel is represented by its central frequency on the y-axis. Each black dot represents the timing of a reaction that was above the detection threshold in each channel. Note that early reactions in low-frequency channels (marked by a yellow arrow) result from the stimulation of those channels caused by the higher frequencies of the downward FM chirp. However, most of these stimulations do not reach the detection threshold and are, therefore, not detected (see Methods). (**C4**) The detections of each channel are convolved with a Gaussian kernel, summed, and compared with the detection threshold (dotted red line). Colored asterisks mark peaks that were classified as **successful detections**—those identified in both the interference-free and full detection conditions (see Methods for details). Other peaks may originate from masking signals or overlapping echoes that did not meet the detection criteria (colors of the sources are as defined above). Panel **D** depicts the pathfinding algorithm used by the bat. The algorithm involves a correlated-random flight during the search phase (blue), collision avoidance (yellow), flying along the wall at a constant distance (green), and flying toward the center of a gap between

*Figure 1 continued on next page*

*Figure 1 continued*

obstacles as an indicator of a possible exit (cyan). After each echolocation call, the bat awaits an IPI (Inter Pulse Interval) period before processing the detections, adjusting flight and echolocation parameters, and emitting the next call. Based on the received signals, it then modifies its next call design and adjusts its direction and speed accordingly. For a detailed diagram of the complete sensorimotor process, see *Figure 1—figure supplement 1*.

The online version of this article includes the following video and figure supplement(s) for figure 1:

**Figure supplement 1.** This diagram illustrates the sensorimotor decision-making process based solely on echolocation.

**Figure supplement 2.** Bat1 receives a reflected echo from Prey1 or a stationary obstacle located at a distance of D from it, with an angle $\phi_{\text{target}}$ relative to its flight direction (red arrow 1).

**Figure 1—video 1.** Example of simulated navigation by 10 bats.

https://elifesciences.org/articles/105571/figures#fig1video1

---

bats/3 m², or 33.3 bats/m²). The bat densities we tested were chosen to reflect the typical range of bat densities observed in natural caves during emergence events (*Goldshtein et al., 2025*; *Fujioka et al., 2021*; *Theriault et al., 2010*). Key model parameters, such as the sensory integration window, object target strength, echolocation parameters, and flight velocity (see *Table 1*), were manipulated and their impact on the exit performance was analyzed. To explicitly quantify the effect of sensory masking vs. the effect of collision avoidance (i.e. spatial interference) only, we turned the acoustic interference on and off to measure its impact. Each scenario was repeated as follows: 1 bat: 240 repeats; 2 bats: 120; 5 bats: 48; 10 bats: 24; 20 bats: 12; 40 bats: 12; 100 bats: 6 (see *Table 1*). The misidentification rate, multi-call clustering, and wall/conspecific target strength analyses (below) were tested only up to 40 bats (see *Table 1*).

## Bats find their way out of the cave even at high conspecific densities

We first examined how bat density affects bats' ability to exit the cave, both alone and in a group. The probability of exiting the cave within 15 s—defined as the proportion of bats that successfully exited within this time frame—was significantly reduced at higher densities (*Figure 2A*, see *Figure 1—video 1* for a view of the bats' movement, $p<10^{-10}$, t = -23, DF = 4077, GLM, see details in *Table 1*). In trials in which a single bat was flying alone, it successfully exited the cave in 100% of the cases. Even without sensory interference, the probability of exiting decreased significantly from 100% to 86±1.4% and 91±1.7% at densities of 100 PKs/3 m² and 100 RMs/3 m², respectively (mean ±s.e.). When acoustic interference was added, the exit probability further decreased to 63±1.4% and 67±1.4% for 100 PKs and RMs, respectively (see *Figure 2A*).

The difference in exit probability between the two species was not significant ($p=0.08$, t=1.74, DF = 4077, GLM as above, *Figure 2A*). Similarly, the difference in echolocation parameters between the two species did not affect the collision rate with the walls, with a maximum of 0.29 and 0.3 collisions per bat per second for PK and RM, respectively, with 100 bats ($p=0.63$, t=−0.48, DF = 4077, GLM, *Figure 2C*, see details in *Table 1*). To quantify sensory interference, we defined a jammed echo as an echo entirely missed due to masking. The jamming probability, which was calculated as the number of jammed echoes divided by the total number of self-echoes, was significantly higher for RM compared to PK. The maximum difference between the two models was 14.3% at a density of 10 bats, with a smaller difference of 9.8% observed at 100 bats ($p<10^{-10}$, t=6.56, DF = 4077, GLM, *Figure 2D*, see details in *Table 1*). Accordingly, PK demonstrated a minor but significant advantage in detecting the cave walls ($p=0.024$, t=−2.25, DF = 4077, GLM, *Figure 2E*, see details in *Table 1*). With 100 bats flying together, the probability of detecting a wall echo at a distance of 1 m in a single call was around 50% and 46% for PK and RM per call, respectively. Despite this minor disadvantage in detection, RM bats exhibited a better time-to-exit average than PK bats, being 0.5 s faster to exit ($p=0.0005$, t=−4.06, DF = 3533, for n=40 bats, *Figure 2B*). Additionally, RM bats experienced a significantly higher probability of their self-generated echoes, reflected off conspecifics, being jammed ($p=0.00016$, t=3.8, DF = 3593, GLM; see details in *Table 1*).

## Multi-call integration improves exit performance

We next examined whether bats improve their performance when integrating information from several consecutive calls. The integration window determines the number of previous calls the bat uses at each step to guide its next movement decision (see Methods and *Figure 3—figure supplement 1*).

**Table 1.** Key model parameters and their effects on performance metrics.

The table presents the key parameters tested, their ranges, default values, and effect sizes on various performance metrics: exit probability, time-to-exit, jamming probability, and collision rate with obstacles. The parameters comprised the number of bats, bat species (PK - *Pipistrellus kuhlii*, RM - *Rhinopoma microphyllum*), integration window, nominal flight speed, call level, echo misidentification with multi-call clustering (yes/no), masking (yes/no), wall target strength, and conspecific target strength. In each scenario, all parameters except the tested one were set to the default value. Call levels are reported in dB-SPL, referenced at 0.1 m from the source. Effect sizes for each parameter are explicitly listed for all four performance metrics, expressed as the change per unit of the tested parameter (e.g. per bat or per 10 dB). For flight speed, a non-monotonic relationship was observed, and values are reported both before and after the peak performance (see Results, *Figure 3B*). Values in square brackets indicate the minimum and maximum of the metric across the tested range. Asterisk (*) indicates a significant impact. Each scenario was tested using Generalized Linear Models (GLMs) with number-of-bats and the tested parameters set as fixed explaining variables. Exit probability and jamming probability were treated as binomially distributed, collision rate was treated as a Poisson distributed, and all other variables were considered normally distributed. Explaining variables were set as fixed factors. The number of repetitions for each scenario was as follows: 1 bat: 240; 2 bats: 120; 5 bats: 48; 10 bats: 24; 20 bats: 12; 40 bats: 12; 100 bats: 6. Ω Misidentification rate, multi-call clustering, wall target strength, and conspecific target strength were simulated only up to 40 bats due to significantly longer run-times. Ψ A significant difference in call intensity was found only for a bat density of 100 bats/3m$^2$, and between the group with a level of 100dB-SPL and all other groups. α see *Figure 3—figure supplement 2*. β see *Figure 3—figure supplement 3*.

| Key parameter | Tested range | Default value | Effect size on explained variable | | | | GLM Explaining factors |
|---|---|---|---|---|---|---|---|
| | | | Exit prob. (%) | Time-to-exit (s) | Jamming- prob. (%) | Obs. Collision (sec$^{-1}$) | |
| Number Of Bats $^Ω$ | [1, 2, 5, 10, 20, 40, 100] | All Values | −0.37/bat* [63:100] | 0.044/bat * [4.6:8.9] | 0.54/bat* [0:54] | 0.25/bat* [0.05:0.3] | Number of bats |
| Bat species | [PK, RM] | PK | 4.5 [63:67] | −0.4* [8.5:8.9] | 9* [54:63] | 0.01 [0.29:0.3]. | Number of bats, bat species |
| Integration Window (#) | [0,1,3,5,10] | 5 | 0.41/call* [29:70] | −0.16/call*[9.2:7.6] | 0.01/call [54:55] | 0.27/call [0.25:0.53] | Number of bats, integration window size |
| Nominal flight speed (m/s) | [2,4,6,8,10] | 6 | 6/ (1m/s),−12/(1 m/s)* [15:63] | −1/(1 m/s),1/ (1m/s)*[6:9.6] | 1.4/ (1m/s)* [55:65] | 0.18/ (1m/s)* [0.07:1.3] | Number of bats, flight speed, square of flight speed |
| Call level (dB-SPL, @ 0.1 m) | [100,110,120,130] | 120 | 13 [50:63] | −0.5 [7.9:8.4] | 0.5/10 dB* [53:58] | 0.07 [0.29:0.36] | Number of bats, call level |
| Misidentification $^Ω$ | [Yes/No] | N | −69* [14:83] | 1.3*[7.6:8.9] | 30* [50:80] | 0.6* [0.2:0.8] | Number of bats, with and without confusion |
| Misidentification and multi-call clustering $^Ω$ | [Yes/No] | N | −23* [58:83] | 1.7*[7.6:9.3] | 29* [50:79] | −0.02* [0.18:0.2] | Number of bats, with and without multi-call clustering |
| Masking | [Yes/No] | Y | 23* [63:86] | 0.8*[8.0:8.8] | 54* [0:54] | 0.03 [0.26:0.29] | Number of bats, with and without masking |
| Wall target strength (dB) $^{α, Ω}$ | [-33,−23, −13,−3] | −23 | 16/10 dB * [23:87] | 0.7/10 dB *[6.6:9.5] | −8.5/10 dB * [34:68] | 0.07/10 dB * [0.19:0.47] | Number of bats, wall target strength |
| Conspecific target strength (dB) $^{β, Ω}$ | [-49,−43, −33,−23] | −23 | −1.5/10 dB [85:91] | 0.25/10 dB*[7.2:8.15] | −0.5/10 dB [48:50] | 0.1/10 dB [0.16:0.2] | Number of bats, conspecific target strength |

In the basic multi-call integration model, detections from the previous calls—by default the last five—were stored in a reference frame, with each detection treated independently as a potential obstacle without clustering or filtering. At each decision, the bat takes all of these detections into account when guiding its movement and echolocation. The probability of exiting the roost significantly increased when increasing the size of the integration window for all bat densities ($p<10^{-10}$, t=28.5, DF = 10197, GLM, *Figure 3A*, see details in *Table 1*). For example, at a density of 40 bats/3 m$^2$, the exit probability improved from 20 to 75% and to 87% as the window size increased from one to three and to 10 previous calls, respectively. In addition, increasing the window size resulted in a significant improvement in the time-to-exit and the avoidance of wall collisions ($p<10^{-10}$, t=−12.8, DF = 7661; $p<10^{-10}$, t=−46.5, DF = 10197, respectively, GLM, see details in *Table 1*). With 100 bats, the collision rate decreased by a factor of 2 from 0.53 to 0.25 collisions per second as the window increased from 1 to 10 calls. The size of the integration window had no significant effect on the jamming probability ($p=0.37$, t=0.9, DF = 10197, GLM, see details in *Table 1*).

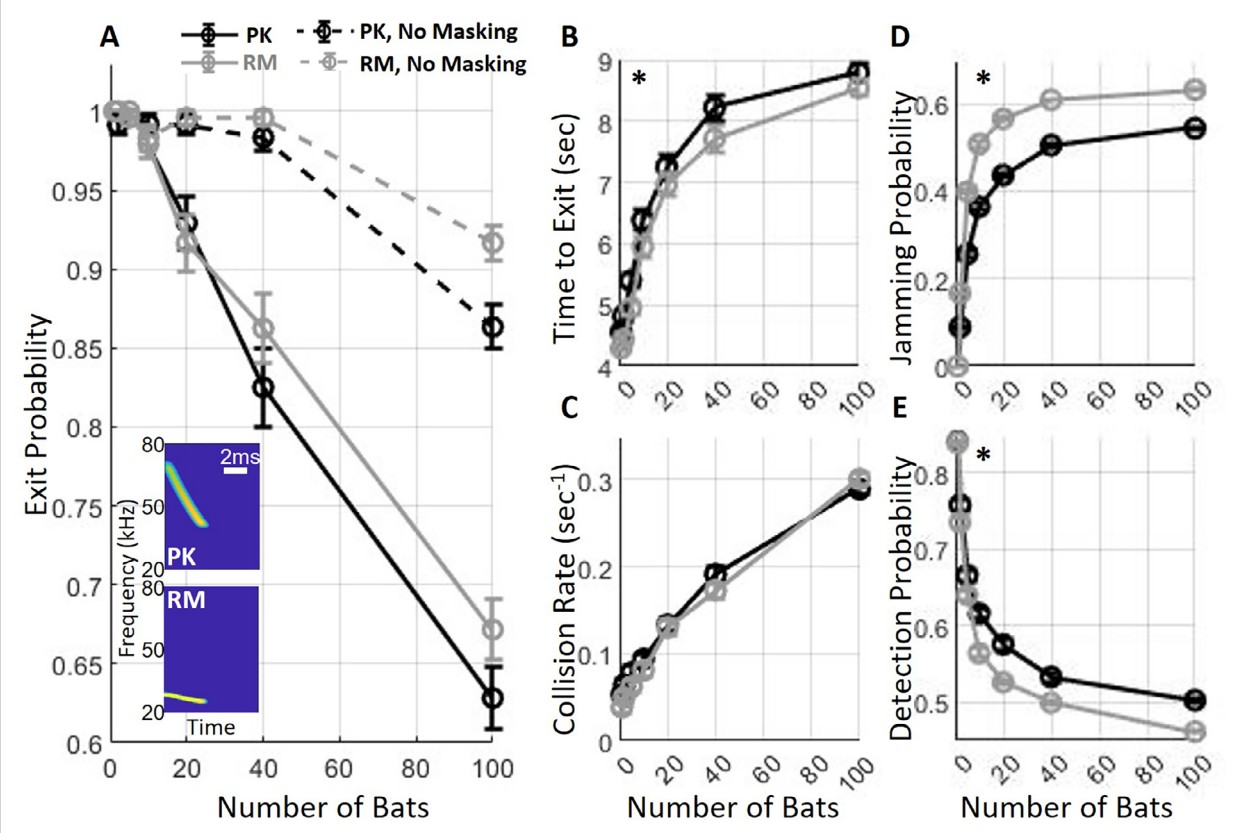

**Figure 2.** Exit performance of *P. Kuhlii* (PK) and *R. Microphyllus* (RM). (**A**) Sensory interference significantly impaired the probability of exiting the cave (compare dashed lines with continuous lines). The probability of a successful exit also declined as the number of bats increased, with no significant difference observed between the species when masking interference was applied. The insert shows the spectrograms of the echolocation calls of PK (top) and RM (bottom). (**B**) The time-to-exit, which was calculated for successful trials only, and (**C**) the collision rate with the walls both increased as a function of the number of bats. (**D**) The probability of jamming significantly increased to about 55 and 63% with 100 bats for PK and RM, respectively. (**E**) The detection probability of a wall reflector at one meter or less in front of a bat decreased as a function of the number of bats. In panels (**A–E**), circles represent means and bars represent standard errors (see details in *Table 1*). Asterisks indicate significant differences between the lines in each panel.

## Exit probability was maximal at an intermediate flight speed

We observed a significant and non-linear effect of the flight speed of the bats on the performance, as shown in *Figure 3B* ($p<10^{-10}$, t=−29.9, DF = 10196, GLM, see details in *Table 1*). The exit probability increased with flight speed until it reached a maximum at 6–8 m/s and then declined rapidly. This was the case for all bat densities, with the maximal exit probability ranging between 65 to 99%. At the optimal velocity, the time to exit was also minimal. However, the collision rate increased monotonically with speed, with a steep incline above the optimal speed.

## Call intensity had only a minor effect on exit performance and only at high bat densities

For low bat densities (<40 bats), call intensity did not have a significant impact on either exit probability or collision rate (*Figure 3C*, $p=0.89$, t=0.13, DF = 5757; $p=82$, t=–0.21, DF = 5757, respectively, GLM, see details in *Table 1*). Call intensity affected exit performance only when the intensity dropped to 100 dB-SPL (@ 0.1 m) and only at a high bat density of 100 bats/3 m$^2$ (*Figure 3C*). In this scenario, the exit probability declined from approximately 60 to 49.5% ($p=0.003$, F=8.45, DF = 2396, One-way ANOVA with 'hsd' post hoc test), and the collision rate increased from 0.3 to 0.35 collisions per second ($p<3×10^{-6}$, F=22.18, DF = 2396). Notably, this low intensity is below the typical search-call intensity of most echolocating bats. At the same bat density (100 bats/3 m$^2$), further increasing the call intensity to above 100 dB-SPL had no significant effect on either exit probability ($p=0.6$) or collision rate ($p=0.07$). Calling louder also slightly, but significantly, decreased the jamming probability at all

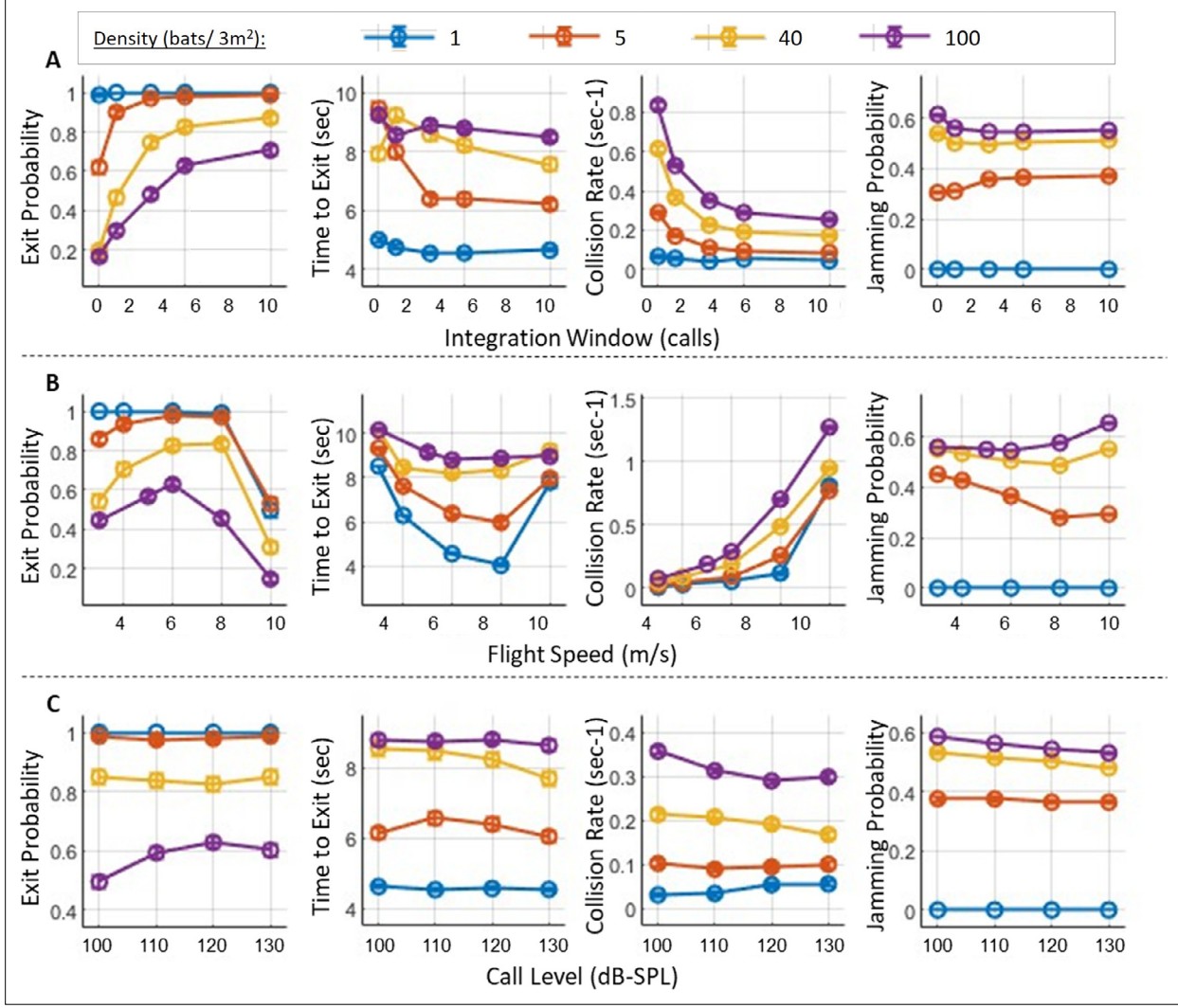

**Figure 3.** Exit performance as a function of key sensorimotor parameters. (**A**) The effect of the integration window on the probability of exiting the roost, the time to exit, the rate of collisions with the walls, and the probability of jamming (from left to right, respectively). Each colored line shows the trend as a function of the window size for different bat densities, with each color representing a specific density. Note that a window size of 0 indicates that only the most recent call is used in the bat's decision-making, without integrating detections from previous calls. (**B**) The effect of the nominal flight speed of the bats, with panels and line colors as in panel A. An optimal speed of approximately 6–8 m/s can be observed for all densities above one bat. (**C**) The effect of call intensity on exit performance, panels as in (**A**). In all panels, circles represent means and bars represent standard errors. Error bars depicting standard errors are presented but are very small due to the large number of simulation repetitions. See *Table 1* for the number of simulated bats.

The online version of this article includes the following figure supplement(s) for figure 3:

**Figure supplement 1.** This figure demonstrates the effect of multi-call integration under non-confusing conditions.

**Figure supplement 2.** This figure shows how changes in the acoustic target strength of the cave walls affect navigation performance across five bat densities (1, 5, 10, 40, and 100 bats/3 m²).

**Figure supplement 3.** This figure shows how changes in the acoustic target strength of conspecifics affect navigation performance across four bat densities (1, 5, 10, and 40 bats/3 m²).

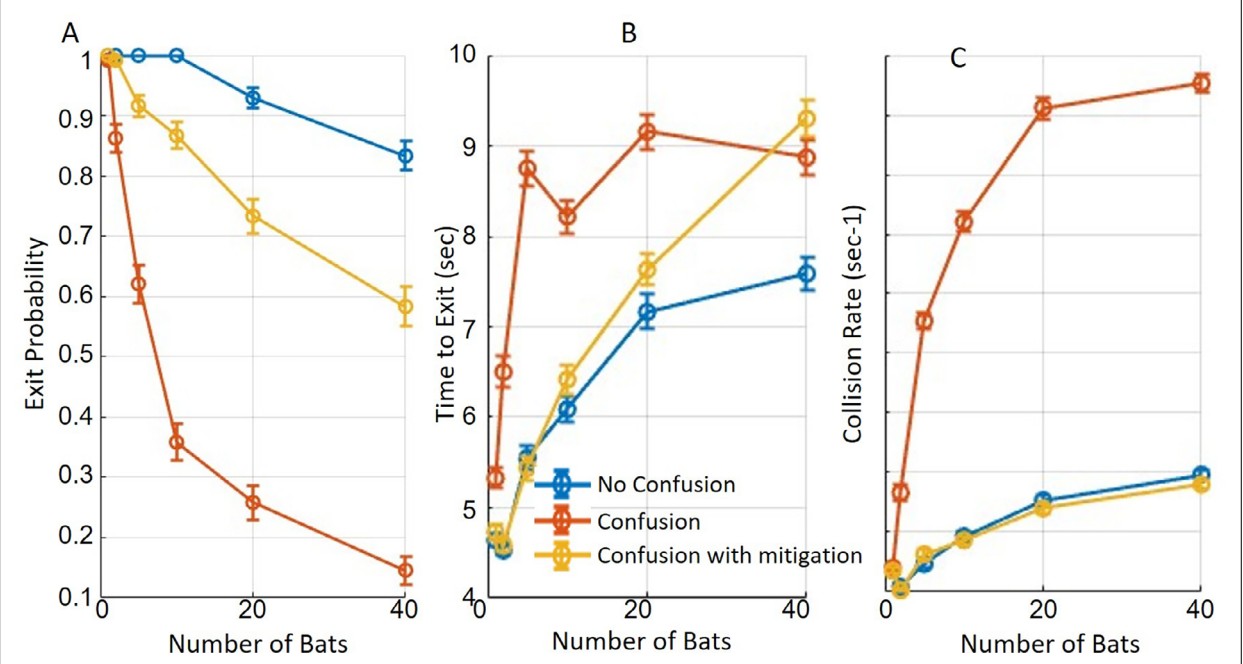

**Figure 4.** The impact of confusion on performance. The figure illustrates the impact of classification confusion on roost-exit performance under various conditions. Blue lines depict trials with masking, while assuming that bats can distinguish between echoes from their own calls and those of conspecifics (referred to as 'No Confusion'). Red lines depict performance where confusion between echoes is assumed. Yellow lines depict performance under the confusion condition, with the added capability of multi-call clustering in a short-term working memory (referred to as 'confusion with mitigation,' see text for further details). In all panels, circles represent means and bars indicate standard errors. (**A**) The probability of exiting the roost significantly decreased with masking and confusion. In conditions with confusion and no aggregation process, only 15% of bats successfully exited the roost, at a density of 40 bats/3 m². Multi-call clustering partially mitigated the confusion effect but did not eliminate it. (**B**) Bats with the ability to distinguish between echoes demonstrated significantly shorter exit times than those experiencing confusion. Note that time to exit refers only to successful attempts. (**C**) The collision rate with walls was highest for bats experiencing both masking and confusion but decreased significantly when without confusion. Multi-call clustering restored performance to the 'No Confusion' condition, reducing collision rates accordingly, at densities between 1–40 bats/3 m².

The online version of this article includes the following figure supplement(s) for figure 4:

**Figure supplement 1.** This figure illustrates the multi-call clustering algorithm under full-confusion conditions.

bat densities, with a decrease of 3.5±8% to 5.5±5% (mean ± s.e.) (*p*=0.02, t=−2.26, DF = 8157, GLM, see *Table 1*).

### While confusion between the desired echoes and those from conspecific calls may significantly impair exit performance, multi-call clustering helps to mitigate this

We next addressed the challenge of echo classification, assuming that a bat can differentiate an echo resulting from its own calls from echoes resulting from the calls of other bats. To examine this assumption, we tested another model, referred to as the **confusion model**, in which bats responded similarly both to wall echoes returning from their own emissions and to those from conspecific emissions, treating all as their own echoes. This confusion significantly decreased exit performance for all bat densities (above one bat). The probability of a successful exit for a density of 40 bats/3 m² dropped from 83.3±2.4%–14.6±2.3% (*p*<<10⁻¹⁰, t=−20.7, DF = 2877, GLM, see details in *Table 1*), the exit time increased from 7.6±0.18 to 9.3±0.2 s (*p*<<10⁻¹⁰, t=15.5, DF = 2157, GLM), and the collision rate increased significantly from 0.2±0.007 to 0.8±0.013 collisions per second (*p*<<10⁻¹⁰, t = -30, DF = 28777, GLM, see *Figure 4*, red and yellow lines).

To further examine whether this substantial decrease in performance could be mitigated even without improving echo identification, we tested an enhanced integration model that, in addition to extending the number of calls integrated, clustered spatially close detections, removed outlier

detections, and estimated wall directions based on grouped reflectors (see Methods and *Figure 4—figure supplement 1*). This **'multi-call clustering'** algorithm significantly improved performance, but exit probability and time-to-exit still remained significantly lower than without echo confusion: exit probability = 58±3% in comparison to 83.3±2.4% without echo confusion ($p \ll 10^{-10}$, t=18.3, DF = 28777, GLM), time-to-exit=9.3±0.2 s ($p \ll 10^{-10}$, t=−13.7, DF = 1996, GLM), see *Figure 4*, yellow line. The results above are reported for a density of 40 bats/3 m². Interestingly, the multi-call clustering restored the collision rate to the levels observed under the 'No Confusion' condition ($p$=0.68, t=−0.42, DF = 2877, GLM, see *Figure 4C*, dark-purple and red lines).

### The effect of wall and conspecific target strengths on exit performance

Increasing the wall target strength significantly enhanced navigation performance (*Figure 3—figure supplement 2*, *Table 1*), improving exit probability by up to 64% and reducing time-to-exit by up to 2.8 s ($p \ll 10^{-10}$). Stronger wall echoes improved wall detection, but also slightly increased masking of desired conspecific signals.

In contrast, changes in conspecific target strength had a much smaller effect (*Figure 3—figure supplement 3*, *Table 1*), with only minor improvements in detection and collision rates and no significant impact on exit probability. This likely reflects the fact that both desired and masking signals scale similarly with conspecific reflectivity. Overall, the model showed low sensitivity to variations in conspecific target strength.

## Discussion

We present a model-based approach that suggests how echolocating bats might find their way out of a crowded roost while contending with severe sensory interference caused by numerous nearby conspecifics. Our results demonstrate that a single bat, lacking prior knowledge of the roost's structure, successfully found the exit in all simulated trials using echolocation alone. As bat density increases, the bats face increased collision risks and more substantial acoustic interference, both of which reduce the probability of efficiently finding the exit. Nevertheless, even at densities of 100 bats/3 m², most bats (63%) successfully exited the roost within a short timeframe. These results are based on a 2D simulation with up to 33 bats/m², under the assumption that bats can distinguish their own echoes from those of conspecifics. We demonstrate how a simple sensorimotor approach can solve this supposedly challenging task. This approach encompasses the following principles: (1) emission of echolocation calls; (2) reception of reflected echoes and masking signals; (3) detection of reflectors (including walls and conspecifics) using a gammatone filter bank biological receiver; (4) localization of the detected objects; (5) employment of multi-call integration of acoustic detections; (6) adjustment of flight and echolocation behavior based on the distance. and angle to the reflectors; and (7) application of simple pathfinding rules to follow walls and gaps while avoiding collisions. Notably, despite the jamming of a substantial percentage of the echoes (with 100 bats - 50% of the echoes from nearby obstacles at ~1 m distance were jammed) the bats managed to maneuver correctly even with this simple approach and partial data.

A key component of this success was the multi-call integration: increasing the number of stored calls from one to ten markedly improved performance, raising the exit probability from 20 to 87% and halving the collision rate. Real bats likely use a much more sophisticated approach that also includes memorizing the roost's structure (*Tsoar et al., 2011*), using landmarks inside the roost (*Jensen et al., 2005*), reliance on the movement of nearby conspecifics (*Couzin et al., 2005*; *Fujioka et al., 2021*), and exploitation of other sensory modalities. We thus expect their actual performance to surpass that of our modeled bats.

Our model suggests that acoustic jamming might be less problematic than has been generally assumed (*Ulanovsky and Moss, 2008*; *Bates et al., 2008*; *Jones and Conner, 2019*), and that movement under severe acoustic masking could be mitigated by increasing the call rate, creating a redundancy across several calls—similar to how real bats behave in a complex environment (*Lin et al., 2016*). In our model, the IPI naturally varied according to established echolocation behavior, decreasing from 100 ms in the search phase to 35 ms (~28 calls per second) in the approach phase, and further to 5 ms (200 calls per second) during the final buzz (*Table 2*). The results indicate that this redundancy, combined with simple sensorimotor heuristics, enhances successful navigation. This

**Table 2.** Echolocation parameters.

The table presents the echolocation parameters of the two bat species we simulated during the specified flight phases (i.e. search, approach, buzz, and final buzz). In each phase, except for the search phase, in which the parameters remain constant, the parameters for each call are determined by the distance to the closest detected object.

*Pipistrellus kuhlii* (Kuhl's pipistrelle)

| Flight phase | Search | Approach | | Buzz | | |
|---|---|---|---|---|---|---|
| Parameter | | Start | End | Terminal 1 start | Terminal 1 end | Terminal 2 |
| Inter Pulse Interval [ms] | 100 | 70 | 35 | 18 | 6 | 5 |
| Call duration [ms] | 7 | 5 | 2 | 2 | 1 | 0.3 |
| Terminal frequency [kHz] | 39 | 39 | 39 | 39 | 39 | 39 |
| Chirp bandwidth [kHz] | 8 | 35 | 30 | 30 | 20 | 20 |
| Call intensity [dB-SPL] | 120 | 120 | 90 | 90 | 80 | 80 |
| Distance to target [m] | >1.2 | 1.2 | 0.4 | 0.4 | 0.2 | <0.2 |

*Rhinopoma microphyllum* (greater mouse-tailed bat)

| Flight phase | Search | Approach | | Buzz | | |
|---|---|---|---|---|---|---|
| Parameter | | Start | End | Terminal 1 start | Terminal 1 end | Terminal 2 |
| Inter Pulse Interval [ms] | 100 | 80 | 20 | 18 | 10 | 9 |
| Call duration [ms] | 12 | 7 | 2 | 2 | 1.5 | 0.75 |
| Terminal frequency [kHz] | 26 | 26 | 26 | 26 | 26 | 23.5 |
| Chirp bandwidth [kHz] | 3 | 4 | 5 | 3 | 3 | 3 |
| Call intensity [dB-SPL, @0.1 m] | 120 | 120 | 90 | 90 | 80 | 80 |
| Distance to target [m] | >1.2 | 1.2 | 0.4 | 0.4 | 0.2 | <0.2 |

is consistent with several recent studies that have pointed in this direction (*Mazar and Yovel, 2020*; *Beleyur and Goerlitz, 2019*; *Goldshtein et al., 2025*).

While echolocation phases—search, approach, and buzz—are traditionally associated with prey capture, similar patterns have been documented in non-foraging tasks, such as landing, obstacle avoidance, clutter navigation, and drinking (*Sabol and Hudson, 1995*; *Betke et al., 2008*; *Tressler and Smotherman, 2009*; *Schnitzler and Kalko, 2001*; *Kalko, 1995*; *Schnitzler et al., 1988*; *Taub and Yovel, 2020*; *Taub et al., 2023*; *Fawcett and Ratcliffe, 2015*; *Grodzinski et al., 2009*; *Yovel et al., 2009*). In these contexts, bats modulate call duration and inter-pulse intervals according to object proximity, generating phase-like transitions even without prey. This supports the interpretation of phase structure as a general proximity-sensing strategy rather than a foraging-specific behavior. In our simulations, bats operated predominantly in the approach phase due to the cluttered cave environment—consistent with natural emergence behavior, where navigation dominates over open-space search. Accordingly, our use of echolocation phases in the model is biologically plausible across a range of sensory-guided tasks.

The bat densities we simulated, ranging from 1 to 100 bats per 3 m², reflect a wide range reported in field studies. Although bat colonies can be much larger than 100 bats, the maximal simulated density in our model (100 bats per 3 m²) resulted in bats flying in very close proximity, with an average nearest-neighbor distance of 0.27 meters. This density is higher than some of the most-dense reported bat aggregations, including studies on Miniopterus fuliginosus (*Fujioka et al., 2021*), Myotis grisescens (*Kazial et al., 2001*), and Tadarida brasiliensis (*Gillam et al., 2010*; *Theriault et al., 2010*; *Kazial et al., 2008*), where bats emerge from the roost at rates of 15–500 bats per second, but fly with an average distance of 0.5 meters between individual bats.

We compared the performance of two FM echolocating insectivorous bat species: Pipistrellus kuhlii (PK) and Rhinopoma microphyllum (RM). PK bats emit wideband echolocation signals that are less prone to jamming than RM bats' narrowband signal (*Cvikel et al., 2015b*; *Yovel et al., 2011*), as wideband signals distribute energy across a broader frequency range and are thus more robust against

interference (*Ulanovsky et al., 2004*; *Schnitzler et al., 2003*). Our findings show that PK signals slightly reduce jamming probability (by 9%) and improve wall detection. However, no significant differences in exit probabilities were noted between the two species.

Using a simulation allowed us to separate the effects of **acoustic interference (masking)** and **spatial interference (collision avoidance)** and revealed new insights into the sensorimotor strategy that could plausibly be used by real bats. The spatial interference reduced the probability of exiting the roost from 100 to 87%, while the acoustic masking further decreased it to 63%. Increasing call intensity had little effect on exit performance, although it slightly improved it at high bat densities. When all bats increased their calling intensity, both desired echoes and masking signals intensified equally, resulting in only a marginal effect. This was tested by varying call intensity levels (100–130 dB SPL) in our simulations (*Table 1*), demonstrating that beyond a certain level (~110 dB SPL), there is no further benefit in improving obstacle detection. These results align with previous studies that have drawn similar conclusions (*Mazar and Yovel, 2020*; *Beleyur and Goerlitz, 2019*).

Bats constantly adjust their flight speed to their surroundings (*Ratcliffe et al., 2004*; *Beetz and Hechavarría, 2022*; *Yovel and Ulanovsky, 2017*; *Moss and Surlykke, 2010*) and specifically when conspecifics are nearby (*Moss and Surlykke, 2001*). Our study suggests that the optimal velocity for flying through a crowded roost ranges from 6 m/s to 8 m/s for densities of 2–100 bats/3 m². Exceeding this velocity range led to a significant drop in exit probability due to a significant increase in wall collisions. We found that this speed did not depend on bat density in accordance with the observations of *Theriault et al., 2010*. Notably, the reported velocities of RM when exiting a cave (*Goldshtein et al., 2025*) and PK emergence velocity near the cave (*Chiu et al., 2009*) are close to the speed that appears optimal, based on our simulations.

We also tested the effects of wall and conspecific target strengths on navigation. Stronger wall echoes substantially improved exit probability and reduced obstacle collisions, despite slightly increasing masking of conspecific echoes (*Figure 3—figure supplement 2*). In contrast, changes in conspecific reflectivity had minimal impact, likely because both desired and masking signals scaled similarly (*Figure 3—figure supplement 3*). This result may also stem from our model's assumption that bats slow down but continue flying at the same direction following a collision with a conspecific.

Our basic model assumed that bats can distinguish **between** wall echoes and conspecific echoes, as **demonstrated** in previous studies (*Salles et al., 2020*; *Kothari et al., 2014*; *Corcoran and Moss, 2017*). We suggest that this is a feasible assumption because echoes from cave walls are longer and exhibit distinct spectro-temporal patterns, whereas echoes from smaller objects, such as conspecifics, are shorter (*Sanderson et al., 2003*; *Munoz and Blumstein, 2012*; *Genzel et al., 2018*). However, wall echoes reflected from conspecific calls might resemble echoes from the bat's own calls in their amplitude and time-frequency characteristics (*Obrist, 1995*; *Moss and Surlykke, 2001*; *Boonman et al., 2013*). This led us to question how the misidentification of such echoes as obstacles might affect navigation. When unable to distinguish between these echoe types, the simulated bats responded to all as if they were their own and thus mis-localized conspecific wall echoes. The confusion led to a substantial drop in exit performance, with only 15% of the bats successfully exiting compared to 82% under no-confusion conditions, at a density of 40 bats/3 m². At the same time, the collision rate increased markedly from 0.2 to 0.85 collisions per second. These results demonstrate the vital importance of echo discrimination for successful navigation, highlighting both the necessity of distinguishing between self and conspecific echoes and the classic challenge of detecting desired signals in noisy environments. There is a substantial evidence in the literature supporting the assumption that bats can recognize their own echoes and reliably distinguish them from those of conspecifics (*Schnitzler et al., 2003*; *Salles et al., 2020*; *Kothari et al., 2014*; *Corcoran and Moss, 2017*; *Danilovich and Yovel, 2019*).

Previous studies have also demonstrated that bats can aggregate acoustic information received sequentially over several echolocation calls, effectively constructing an auditory scene in complex environments (*Ulanovsky and Moss, 2008*; *Harten et al., 2020*; *Prat and Yovel, 2020*; *Barchi et al., 2013*; *Geva-Sagiv et al., 2015*; *Gunier and Elder, 1971*). Bats are also known to emit call sequences in groups, particularly when spatiotemporal localization demands are high. Studies have recorded sequences of 2–15 grouped calls, supporting the idea that grouping facilitates echo aggregation (*Prat and Yovel, 2020*; *Garnier et al., 2007*). Accordingly, we tested how multi-call clustering process— which included grouping nearby reflectors, removing outliers, and estimating wall orientation based

on these clusters—could assist bats in pathfinding, even under the assumption of full confusion. At bat densities of 1–40 bats/3 m² with masking, the multi-call clustering completely restored the collision rate with walls from 0.85 back to 0.2 collisions per second and significantly improved the exit probability, raising it to 58%, although it did not entirely eliminate the impact of confusion. Our assumption of total confusion between echoes from a bat's own calls and those from conspecifics, as well as our relatively simple clustering model, likely underestimates the true capabilities of real bats when flying in complex environments.

Navigation in bats involves processing complex sensory inputs and applying effective decision-making, often requiring an ability to switch strategies (*Fischer et al., 2018*; *Schnitzler et al., 1987*; *Wilson and Moss, 2004*; *Kober and Schnitzler, 1990*; *Kuc, 1994*; *Vanderelst and Peremans, 2018*; *Simmons and Kick, 1983*). Bats possess a highly accurate spatial memory (*Harten et al., 2020*; *Wilson and Moss, 2004*; *Simmons and Kick, 1983*; *Griffin et al., 1960*; *Hiryu et al., 2008*), which is essential for tasks like long-distance migration (*Tsoar et al., 2011*), homing (*Hagino et al., 2007*), and maneuvering in cluttered environments (*Griffin et al., 1960*). Additionally, they utilize acoustic landmarks to orient in total darkness (*Jensen et al., 2005*), occasionally rely on vision (*Kober and Schnitzler, 1990*; *Kuc, 1994*), particularly at the cave edge where light is available, can passively detect echolocating peers, and perhaps eavesdrop on conspecifics' echoes (*Lin and Abaid, 2015*). In this study, we focused on whether echolocation alone is sufficient for one of the most difficult orientation tasks that bats perform–exiting a roost at high densities without prior knowledge of the roost's shape, aside from the initial flight direction. Thus, our echolocation-only model, which was based on a five-call integration window during most simulations, probably underestimates real bats' actual performance, which also benefits from additional sensory input and can employ additional navigation strategies by sharing information between each other to coordinate and optimize the routes, such as manifested by swarming intelligence (*Youssefi and Rouhani, 2021*; *Surlykke et al., 2009*; *Ghose et al., 2006*).

Our model highlights the importance of considering sensory interference in animal behavior research and illuminates the impressive capabilities of echolocating bats. Additionally, the model showcases the value of simulations and establishes a framework for future studies on collective movement and swarming animals, and on robotics in complex environments.

## Methods

The simulated bats rely solely on echolocation to detect and locate obstacles and other bats by analyzing the sound waves they receive. They emit directional echolocation calls and receive the echoes reflected by roost walls and conspecifics, as well as the calls of conspecifics and the echoes returning from their calls. The bats adjust their flight trajectory and echolocation behavior based on the estimated location of the detected objects (range and angle), which deteriorates upon acoustic interference. The detection of the received signals is based on the mammalian gammatone filter bank receiver, under the assumption that bats can differentiate between the desired detected obstacles, conspecifics' echoes, and masking signals. We conducted 2D simulations with varying numbers of bats (from 1 to 100) to analyze the flight trajectories with and without masking interference by conspecifics. In the trials without masking interference, the bats successfully detected walls and conspecifics without any hindrance. While real-world bat navigation occurs in 3D space, the 2D framework represents a worst-case scenario for echolocation-based navigation, as it increases effective bat density and limits maneuverability compared to a full 3D environment. This approach provides a conservative test of jamming and collision avoidance while maintaining computational tractability, allowing for extensive simulation runs to explore different variables systematically. For a detailed description of the MATLAB simulation, see *Mazar and Yovel, 2020*.

The simulation arena was designed to mimic a roost with a corridor-like layout, measuring 14.5 meters in length and 2.5 meters in width, featuring a right-angle turn located 5.5 meters before the exit (see *Figure 1A* for a top-down view). All bats started at a random position within a 2×1.5 m area at the far end of the cave, each initiating flight within a 0.1 s window in a random direction between –30° and +30° relative to the exit (see *Figure 1*). They employ a simple navigation algorithm that dynamically adjusts flight direction based on the detected obstacles or conspecifics (*Figure 1D*, *Figure 1—figure supplement 1*). If no obstacles or conspecifics are detected, they continue in a correlated random walk with a maximal turning rate of approximately 30 deg/s. When obstacles are detected, they are first localized with an error (see below and *Mazar and Yovel, 2020*). Then, if an

opening (i.e. a gap of at least 0.5 m between obstacles) is detected, the bats fly through it; if not, they follow the walls while maintaining a 0.8 m distance from them. When approaching an obstacle too closely (<1.5 m and at an angle <60°), they execute an obstacle avoidance maneuver. Close proximity to another bat (<0.4 m) triggers an avoidance maneuver away from the nearest conspecific. To evaluate the choice of these distances (1.5 m from walls and 0.4 m from other bats), we tested the sensitivity of the model to conspecific avoidance distances ranging from 0.2 to 1.6 meters across bat densities of 2–40 bats/3 m². We observed only a modest effect on exit probability at the highest density, where exit probability increased slightly from 82 to 88% (*p*=0.024, t=2.25, DF = 958). No significant changes were observed in exit time, collision rate, or jamming probability across other densities or conditions (GLM, with the number of bats and avoidance distance set as fixed explanatory variables, and the outcome variable being one of: exit probability, time-to-exit, collision rate, or jamming probability). These findings confirm that the modeled behavior is largely insensitive to this parameter range.

If the bat collides with a wall, it immediately turns so that its new flight direction is at a 90° angle to the wall. Collisions between conspecifics, which are common in nature and generally not disruptive in low velocities, are not explicitly modeled. Instead, during the collision event, the bat keeps decreasing its velocity and changing its flight direction until the distance between bats is above the threshold (0.4 m). We assume that the primary cost of such interactions arises from the effort required to avoid collisions resulting in forced changes in flight direction and speed, rather than from the collision itself. Each decision relies on a multi-call integration window that records the estimated locations of detected reflectors from recent echolocation calls (see *Figure 3—figure supplement 1*). By default, this window includes the last five calls, and we systematically tested the effect of using between 1 and 10 calls. This algorithm functions without any prior knowledge of the bats' location or the roost's structure. To assess performance, we measured the probability of successfully exiting the roost within a 15 s window. The time-based exit limit was chosen because it is approximately twice the average exit time for 40 bats under acoustic interference in our model, allowing bats sufficient time to correct their trajectory and circle back if they missed the exit on the first attempt. This threshold keeps simulation times reasonable while still capturing the key aspects of exit dynamics.

Echolocation behavior and flight speed follow the phases widely reported in insectivorous bats, categorized as 'search,' 'approach,' and 'buzz' (*Betke et al., 2008*; *Mazar, 2016*; *Friis, 1946*; *Rahman et al., 2015*; *Boonman et al., 2003*; *Boonman and Ostwald, 2007*) with specific echolocation parameters for Pipistrellus kuhlii (Kuhl's pipistrelle) (*Beetz and Hechavarría, 2022*) and Rhinopoma microphyllum (greater mouse-tailed bat, *Goldshtein et al., 2025*). The transition distances between these phases were identical for both species (see *Table 2*) and are based on empirical studies documenting hunting and obstacle avoidance behavior (*Betke et al., 2008*; *Tressler and Smotherman, 2009*; *Ratcliffe et al., 2004*; *Boonman et al., 2003*; *Boonman and Ostwald, 2007*; *Weissenbacher and Wiegrebe, 2003*). In nature, call parameters (IPI, call duration, and start and stop frequencies) are primarily shaped by the target distance and echo strength. Accordingly, there is little difference in echolocation between prey capture and obstacles-related maneuvers, aside from intensity adjustments based on target strength (*Tressler and Smotherman, 2009*; *Schnitzler and Kalko, 2001*; *Garnier et al., 2007*; *Saillant et al., 1993*). In our study, due to the dense cave environment, the bats are found to operate in the approach phase nearly all of the time, which is consistent with natural cave emergence behavior, where they are navigating through a cluttered environment rather than engaging in open-space search. Our model was designed to remain as simple as possible while relying on conservative assumptions that may underestimate bat performance. If, in reality, bats fine-tune their echolocation calls even earlier or more precisely during navigation than assumed, our model would still conservatively reflect their actual capabilities.

The simulated echolocation call consists of the dominant harmony of the bat's FM chirp (first harmony of the PK and second harmonies of the RM). The echolocation signals used in our simulation were modeled as logarithmic FM chirps, implemented using the MATLAB built-in function (e.g. chirp (t, f0, t1, f1, 'logarithmic')). This approach aligns with the known nonlinear frequency modulation characteristics of PK and RM. *Table 2* provides the specific echolocation parameters used in the model, based on *Kalko, 1995*, *Ratcliffe et al., 2004*, and *Goldshtein et al., 2025*. During the search phase, the bats fly at a nominal velocity of 6 m/s, reducing it by half during the approach phase and continuously adjusting their speed according to the relative direction of the target, using a delayed

linear adaptive law (*Mazar and Yovel, 2020*; *Boonman et al., 2003*; *Popper and Fay, 1995*). The manoeuvrability of the bats is constrained to a maximum of 4 m/s², limiting both angular and linear accelerations. Additionally, our model includes random individual variations in terminal frequencies, assuming a normal distribution with a standard deviation of 1 kHz across the bats.

The sound intensity of the echoes generated by the bat's own calls and those of its conspecifics are calculated using the sonar equation (*Mazar and Yovel, 2020*; *Blauert, 1996*) (pp. 196–198), as shown in *Equation 1*, geometrical relations are according to *Figure 1—figure supplement 2*. The received levels of the masking calls are determined by using the Friis transmission equation (*Mohl and Surlykke, 1989*), as shown in *Equation 2*. All signal levels were simulated and reported in dB-SPL, referenced to 0.1 meters from the emitting bat. Bats are modeled acoustically as spherical reflectors with a fixed target strength of –23 dB assuming reference distance 1 m, reflecting sound isotropically. This approximates a sphere with a radius of 0.15 m, corresponding to the approximate wingspan of RM (*Goldshtein et al., 2025*; *Simmons et al., 1983*). While target strength can vary with wing posture and body geometry, we chose a representative value within the reported biological range for simplicity and model consistency. Our own measurement of a 3D-printed RM bat yielded a target strength of –32 dB, and a sensitivity analysis showed that model performance was only mildly affected across a wide range of target strengths (see *Figure 3—figure supplement 3*). This supports the robustness of our approach to different sized bats. Walls are modeled as composites of individual reflectors placed 20 cm apart; each treated as a sphere with a 20 cm radius and a target strength of –22.5 dB. For simplicity, in our model, the head is aligned with the body; therefore,, the direction of the echolocation beam is the same as the direction of the flight. The directivity of the calls and the received echoes is defined by the piston model (*Mazar and Yovel, 2020*; *Rahman et al., 2015*) with radii of 3 mm for the mouth gap and 7 mm for the ear. The directivity is not directly influenced by velocity but follows behavioral-dependent frequency changes. As the bat transitions from search to approach to buzz phases, it emits higher-frequency calls, leading to increased directivity. This shift coincides with a natural reduction in speed during the approach phase. Echo delays are calculated as the two-way travel time of the signals from the emitter to the target.

$$P_r = P_t \cdot \frac{G_t\left(\phi_{target}, f\right) \cdot G_r\left(\phi_{target}, f\right) \lambda^2}{(4\pi)^3 D^4} \cdot 10^{-2\alpha_{att}(f)/10 \cdot (D-0.1)} \cdot \sigma(f) \tag{1}$$

$$P_{mask} = P_t G_t\left(\phi_{t_x r_x}, f\right) G_r\left(\phi_{r_x t_x}, f\right) \cdot \left(\frac{\lambda}{4\pi D_{txrx}}\right)^2 10^{-\alpha_{att} \cdot (D-0.1)} \tag{2}$$

$$P_{echoesFromMasking} = P_t \cdot \frac{G_t\left(\phi_{t_x}, f\right) \cdot G_r\left(\phi_{r_x}, f\right) \lambda^2}{(4\pi)^3 D_{t_x}^2 D_{r_x}^2} \cdot 10^{-\alpha_{att} \cdot (D_{t_x} + D_{r_x} - 0.2)} \cdot \sigma(f) \tag{3}$$

where,

$G_r(\phi, f)$ : level of the received signal [SPL]

$P_t$: level of the transmitted call [SPL]

$P_{mask}$ : level of the masking signal as received by the bat [SPL]

$P_{echoesFromMasking}$: level of the echoes reflected by conspecifics and received by the bat [SPL]

$G_t(\phi, f)$: gain of the transmitter (mouth of the bat, piston model), as a function of azimuth and frequency (f) [numeric]

$G_r(\phi, f)$ : gain of the receiver (ears of the bat, piston model) [numeric]

$\phi_{target}$ : the angle between the bat and the reflected object [rad]

D : distance between the bat and the target [m]

$\phi_{t_x r_x}, D_{t_x r_x}$ : the angle [rad], and the distance [m] between the transmitting conspecific and the receiving focal bat (from the transmitter's perspective)

$\phi_{r_x t_x}, D_{r_x t_x}$ : the angle [rad], and the distance [m] between the receiving bat and the transmitting bat (from the receiver's perspective)

$\phi_{t_x}$ : the angle [rad], between the masking bat and target (from the transmitter's perspective)

$\alpha_{att}(f)$ : atmospheric absorption coefficient for sound [dB/m]

$\sigma(f)$ : SONAR cross-section of the target [m²]

$\lambda$ : The wavelength of the signal [m]

To maintain model simplicity, we did not incorporate Doppler effects in the echolocation model. While Doppler shifts can affect frequency perception, their impact on jamming and navigation performance is minimal in this context (*Popper and Fay, 1995*). Moreover, the inter-individual random signals frequencies were larger than the expected Dopplers. In addition, the model does not assign echoes to earlier calls if their delays exceed the bat's own IPI and thus does not simulate pulse-echo ambiguity.

To model the detection process in the bat's cochlea, we employed a monoaural filter bank receiver (*Sanderson et al., 2003*, *Boonman and Ostwald, 2007*, *Weissenbacher and Wiegrebe, 2003*) consisting of 80 channels, each with three components: (i) a gammatone filter of order 8, acting as a bandpass filter with center frequencies logarithmically scaled between 10 kHz and 80 kHz (*Mazar and Yovel, 2020*); (ii) a half-wave rectifier; and (iii) a low-pass filter (Butterworth, fc = 8 kHz, order = 6). Object detection and distance estimation are conducted using Saillant's method (*Mazar and Yovel, 2020*; *Sanderson et al., 2003*, *Saillant et al., 1993*), based on the sum of detections in the active channels, see *Figure 1C and D*. Initially, a de-chirping process calculates the reference frequency-delay by detecting the peak in the response of each channel to the emitted call in a noise-free environment. Subsequently, the received signal, containing both desired echoes and masking sounds, passes through the filter bank. In each channel, all peaks above a threshold level are detected and time-shifted by the de-chirp reference. The detection threshold in each channel was set to the higher of two values: either 7 dB above the noise floor (0 dB-SPL) or the maximum received signal level minus 70 dB, thereby enforcing a dynamic range of 70 dB. Peaks from all channels are aggregated in 5 µs windows and convolved with a Gaussian kernel with σ=5 µs. Output peaks that exceed the threshold level, set at 10% of the number of active channels, and fall within a time window of 100 µs around the expected delay are considered successful detections.

To evaluate the impact of acoustic interference, we conducted the detection procedure twice. The first, termed 'interference-free detection,' comprised only the desired echoes, with white Gaussian noise at a level of 0 dB-SPL and without masking signals. The second, termed 'full detection' comprised the desired echoes, Gaussian noise, and the masking signals. Detected echoes in the full detection were defined by the strongest peak within a 4 ms window 3 ms before and 1 ms after, accounting for forward and backward masking (*Beleyur and Goerlitz, 2019*, *Popper and Fay, 1995*; *Blauert, 1996*; *Mohl and Surlykke, 1989*) detected above the threshold within 100 µs of the interference-free detections. If the detected peak in the full detection condition was delayed by more than 100 µs compared to the interference-free case, it was defined as a miss-detection. Peaks with smaller timing shifts were considered **detections with timing errors. Jammed echoes** were defined as echoes that were detected under the interference-free condition but not detected under the full detection condition. The **jamming probability** was calculated as the ratio of jammed echoes in the full detection condition to the detected echoes in the interference-free condition.

After detection, the bat estimates the range and the Direction of Arrival (DOA) of the reflecting objects. The range is determined by the delay of the detected echo, including any errors derived from the filter-bank process in the 'full detection' process (i.e. including all masking signals) (*Mazar and Yovel, 2020*, *Simmons et al., 1983*, *Weissenbacher and Wiegrebe, 2003*). The direction is not explicitly estimated through binaural processing. Instead, based on previous studies (*Simmons et al., 1983*; *Popper and Fay, 1995*), we assumed that bats can estimate the direction of arrival with an angular error that depends on the Signal-to-Noise Ratio (SNR) and the angle. The inputs to this process include the peak level of the desired echo, the noise level, and the level of acoustic interference. The output is the estimated direction of arrival with a random error applied based on the SNR. At an angle of 0° and an SNR of 10 dB, the standard deviation of the error is 1.5° (*Popper and Fay, 1995*) and (*Mazar and Yovel, 2020*; *Equation 4*), with the error capped at a maximum of 3° in our model.

$$DOA_{error} = \sqrt{\left(\frac{k_2}{SNR}\right)^2 + \left(k_3 + k_4 \cdot \sin\left(\phi\right)\right)^2} \qquad (4)$$

where $k_2$, $k_3$, and $k_4$ are constants chosen to produce a DOA error consistent with the range described above.

To evaluate the impact of the assumption that bats can distinguish between echoes caused by their own calls and those caused by other bats (i.e. conspecifics' reflectors), we tested an alternative model

in which the simulated bats treat all echoes reflected from walls as if they have originated from their own calls. The distance to reflectors of conspecifics' calls is estimated based on the time difference between the echo and the bat's last call. The direction of arrival is estimated by the angle between the bat and the physical reflector, with an added random error (the same process used for their own echoes).

In real bats, spatial processing in the brain involves integrating auditory and spatial information over time to construct a coherent map of the environment (*Ulanovsky and Moss, 2008*; *Schnitzler et al., 2003*). This neural computation is crucial for navigation and prey detection in complex environments. To examine whether spatial integration mitigates the confusion problem, we added a 'multi-call clustering' module that was based on the sensory information obtained within a 1 s memory window. The clustering comprised the following steps: (i) clustering all detections in memory into groups with a maximum internal distance of 10 cm; (ii) reconstructing the estimated walls positions and directions based on the average of clusters that include at least two detections (rather than relying on single reflections); and (iii) identifying openings between reconstructed wall edges ranging from 0.5 to 2.25 m in width, see *Figure 1—figure supplement 1*, *Figure 4—figure supplement 1*. The model assumes that bats store echo locations in an allocentric x-y coordinate system, transforming detections from a local to a global spatial framework. Collision avoidance is based not only on the integrated spatial representation but also on immediate echoes from the last call (prior to clustering), including potential uncorrected false detections and localization errors, which are independently processed for real-time evasive maneuvers.

## Statistical analysis

Statistical analysis and the roost-exit model were conducted using MATLAB 2023 a.

Tests were performed with a significance level of 0.05. For each simulated scenario, we examined the effect of the various parameters on exit probability, time-to-exit, collision rate, and the jamming probability, using Generalized Linear Models (GLMs). The GLM tests were executed with MATLAB built-in function **'fitglm()'**. Probability variables (such as exit and jamming probabilities) were treated as binomially distributed; rate variables (such as collision rate) were treated as Poisson distributed, and all other variables were considered normally distributed. Unless otherwise stated, all explaining variables were set as fixed factors. All statistical analyses, including the statistical test and the corresponding sample sizes, are described throughout the text and summarized in *Table 1*. Standard errors are calculated across all individuals within each scenario, without distinguishing between different simulation trials.

---

## Additional information

### Funding

| Funder | Grant reference number | Author |
|---|---|---|
| Israel Science Foundation | 331/22 | Omer Mazar<br>Yossi Yovel |

The funders had no role in study design, data collection and interpretation, or the decision to submit the work for publication.

### Author contributions

Omer Mazar, Conceptualization, Software, Formal analysis, Investigation, Visualization, Methodology, Writing – original draft, Writing – review and editing; Yossi Yovel, Conceptualization, Supervision, Funding acquisition, Validation, Methodology, Project administration, Writing – review and editing

### Author ORCIDs

Omer Mazar ⓘ https://orcid.org/0000-0001-9763-4621
Yossi Yovel ⓘ https://orcid.org/0000-0001-5429-9245

Reviewer #1 (Public review): https://doi.org/10.7554/eLife.105571.4.sa1

Reviewer #2 (Public review): https://doi.org/10.7554/eLife.105571.4.sa2
Author response https://doi.org/10.7554/eLife.105571.4.sa3

## Additional files

### Supplementary files
MDAR checklist

### Data availability
All data and codes generated during this study are included in the manuscript and supporting files. Source code files have been uploaded with a Graphical User Interface and a readme file for explanation. Data are available at Zenodo and Github: https://doi.org/10.5281/zenodo.16992617, https://github.com/omermazar/Colony-Exit-Bat-Simulation, copy archived at *Mazar, 2025*.

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
