## [Editor Report · eLife Assessment]

This **important** model-based study seeks to mimic bat echolocation behavior and flight under conditions of high interference, such as when large numbers of bats leave their roost together. The simulations **convincingly** suggest that the problem of acoustic jamming in these situations may be less severe than previously thought. This finding will be of broad interest to scientists working in the fields of bat biology and collective behaviour.

---

## [Referee Report · Reviewer #1 (Public review)]

Summary:

Mazer & Yovel 2025 dissect the inverse problem of how echolocators in groups manage to navigate their surroundings despite intense jamming using computational simulations.

The authors show that despite the 'noisy' sensory environments that echolocating groups present, agents can still access some amount of echo-related information and use it to navigate their local environment. It is known that echolocating bats have strong small and large-scale spatial memory that plays an important role for individuals. The results from this paper also point to the potential importance of an even lower-level, short-term role of memory in the form of echo 'integration' across multiple calls, despite the unpredictability of echo detection in groups. The paper generates a useful basis to think about the mechanisms in echolocating groups for experimental investigations too.

Strengths:

The paper builds on biologically well-motivated and parametrised 2D acoustics and sensory simulation setup to investigate the various key parameters of interest

The 'null-model' of echolocators not being able to tell apart objects & conspecifics while echolocating still shows agents successfully emerge from groups - even though the probability of emergence drops severely in comparison to cognitively more 'capable' agents. This is nonetheless an important result showing the direction-of-arrival of a sound itself is the 'minimum' set of ingredients needed for echolocators navigating their environment.

The results generate an important basis in unraveling how agents may navigate in sensorially noisy environments with a lot of irrelevant and very few relevant cues.

The 2D simulation framework is simple and computationally tractable enough to perform multiple runs to investigate many variables - while also remaining true to the aim of the investigation.

---

## [Referee Report · Reviewer #2 (Public review)]

This manuscript describes a detailed model for bats flying together through a fixed geometry. The model considers elements which are faithful to both bat biosonar production and reception and the acoustics governing how sound moves in air and interacts with obstacles. The model also incorporates behavioral patterns observed in bats, like one-dimensional feature following and temporal integration of cognitive maps. From a simulation study of the model and comparison of the results with the literature, the authors gain insight into how often bats may experience destructive interference of their acoustic signals and those of their peers, and how much such interference may actually negatively effect the groups' ability to navigate effectively. The authors use generalized linear models to test the significance of the effects they observe.

The work relies on a thoughtful and detailed model which faithfully incorporates salient features, such as acoustic elements like the filter for a biological receiver and temporal aggregation as a kind of memory in the system. At the same time, the authors abstract features that are complicating without being expected to give additional insights, as can be seen in the choice of a two-dimensional rather than three-dimensional system. I thought that the level of abstraction in the model was perfect, enough to demonstrate their results without needless details. The results are compelling and interesting, and the authors do a great job discussing them in the context of the biological literature.

With respect to the first version of the manuscript, the authors have remedied all my outstanding questions or concerns in the current version. The new supplementary figure 5 is especially helpful in understanding the geometry.

---

## [Author Response]

The following is the authors’ response to the previous reviews

**Public Reviews:**

**Reviewer #1 (Public review):**
Summary:Mazar & Yovel 2025 dissect the inverse problem of how echolocators in groups manage to navigate their surroundings despite intense jamming using computational simulations.The authors show that despite the 'noisy' sensory environments that echolocating groups present, agents can still access some amount of echo-related information and use it to navigate their local environment. It is known that echolocating bats have strong small and large-scale spatial memory that plays an important role for individuals. The results from this paper also point to the potential importance of an even lower-level, short-term role of memory in the form of echo 'integration' across multiple calls, despite the unpredictability of echo detection in groups. The paper generates a useful basis to think about the mechanisms in echolocating groups for experimental investigations too.Strengths:The paper builds on biologically well-motivated and parametrised 2D acoustics and sensory simulation setup to investigate the various key parameters of interestThe 'null-model' of echolocators not being able to tell apart objects & conspecifics while echolocating still shows agents succesfully emerge from groups - even though the probability of emergence drops severely in comparison to cognitively more 'capable' agents. This is nonetheless an important result showing the direction-of-arrival of a sound itself is the 'minimum' set of ingredients needed for echolocators navigating their environment.The results generate an important basis in unraveling how agents may navigate in sensorially noisy environments with a lot of irrelevant and very few relevant cues.The 2D simulation framework is simple and computationally tractable enough to perform multiple runs to investigate many variables - while also remaining true to the aim of the investigation.Weaknesses:Authors have not yet provided convincing justification for the use of different echolocation phases during emergence and in cave behaviour. In the previous modelling paper cited for the details - here the bat-agents are performing a foraging task, and so the switch in echolocation phases is understandable. While flying with conspecifics, the lab's previous paper has shown what they call a 'clutter response' - but this is not necessarily the same as going into a 'buzz'-type call behaviour. As pointed out by another reviewer - the results of the simulations may hinge on the fact that bats are showing this echolocation phase-switching, and thus improving their echo-detection. This is not necessarily a major flaw - but something for readers to consider in light of the sparse experimental evidence at hand currently.

The use of echolocation phases—defined as the sequential search, approach, and buzz call patterns—has been documented not only during foraging but also in tasks such as landing, obstacle avoidance, clutter navigation, and drinking. Bat call structure has been shown to vary systematically with object proximity, not exclusively in response to prey. During obstacle avoidance, phase transitions were observed, with approach calls emitted in grouped sequences and with reduced durations (Gustafson & Schnitzler, 1979; Schnitzler et al., 1987). In landing contexts, bats have been reported to emit short-duration calls and decrease inter-pulse intervals—buzz-like patterns also observed during prey capture— suggesting shared acoustic strategies across behaviors (Hagino et al., 2007; Hiryu et al., 2008; Melcón et al., 2007, 2009). Comparable patterns have been reported during drinking maneuvers, where “drinking buzzes” have been proposed to guide a precise approach to the water surface, analogous to landing buzzes (Griffiths, 2013; Russo et al., 2016). In response to environmental complexity, bats were found to shorten calls and increase repetition rates when navigating cluttered spaces compared to open ones (Falk et al., 2014; Kalko & Schnitzler, 1993).

Moreover, field recordings from our study of Rhinopoma microphyllum (Goldshtein et al., 2025) revealed shortened call durations and inter-pulse intervals during dense group flight outside the cave during emergence—patterns consistent with terminal-approach phase that is typical when coming very close to an object (another bat in this case). The Author response image 1 shows an approach sequence recorded from a tagged bat approximately 20 meters from the cave entrance, with self-generated echolocation calls marked. The inter-pulse-interval of ca. 20 ms is used by these bats when a reflective object (another bat in this case) is nearby.

These results provide direct evidence that bats actively employ approach-phase echolocation during swarming likely to avoid collision with other bats. This supports the view that echolocation phase transitions are a general proximity-based sensing strategy, adapted across a variety of behavioral scenarios—not limited to hunting alone.

In our simulations, bats predominantly emitted calls in the approach phase, with only rare occurrences of buzz-phase calls.

See lines 355-363 in the revised manuscript.

The decision to model direction-of-arrival with such high angular resolution (1-2 degrees) is not entirely justifiable - and the authors may wish to do simulation runs with lower angular resolution. Past experimental paradigms haven't really separated out target-strength as a confounding factor for angular resolution (e.g. see the cited Simmons et al. 1983 paper). Moreover, to this reviewer's reading of the cited paper - it is not entirely clear how this experiment provides source-data to support the DoA-SNR parametrisation in this manuscript. The cited paper has two array-configurations, both of which are measured to have similar received levels upon ensonification. A relationship between angular resolution and signal-to-noise ratio is understandable perhaps - and one can formulate such a relationship, but here the reviewer asks that the origin/justification be made clear. On an independent line, also see the recent contrasting results of Geberl, Kugler, Wiegrebe 2019 (Curr. Biol.) - who suggest even poorer angular resolution in echolocation.

We thank the reviewer for raising this important point. The acuity of 1.5–3° in horizontal direction-of-arrival (DoA) estimation is based on the classical work of Simmons et al. with *Eptesicus fuscus* (Simmons et al., 1983). Similar precision was later supported by Erwin et al. (Erwin et al., 2001), who modeled azimuth estimation from measured interaural intensity differences (IIDs), reporting an average error of 0.2° with a standard deviation of ~2.2°, consistent with the behavioral data found by Simmons. The decline in acuity with increasing arrival angle has also been demonstrated in behavioral and physiological studies of binaural IID processing (Erwin et al., 2001; Fay, 1995; Razak, 2012; Wohlgemuth et al., 2016). The error model itself was first introduced in our earlier work (Mazar & Yovel, 2020).

Importantly, Geberl et al. (Geberl et al., 2019) examined the resolution of weak targets masked by nearby strong flankers and found poor spatial discrimination of ~45 degrees; however, they were studying a detection problem, rather than the horizontal acuity of azimuth estimation. Indeed, our model assumes there is no spatial discrimination at all.

Overall, while our DoA–SNR parametrization can certainly be critiqued and alternative parameterizations could be tested in future work, we believe it reflects a reasonable and empirically supported assumption.

**Reviewer #2 (Public review):**
This manuscript describes a detailed model for bats flying together through a fixed geometry. The model considers elements which are faithful to both bat biosonar production and reception and the acoustics governing how sound moves in air and interacts with obstacles. The model also incorporates behavioral patterns observed in bats, like one-dimensional feature following and temporal integration of cognitive maps. From a simulation study of the model and comparison of the results with the literature, the authors gain insight into how often bats may experience destructive interference of their acoustic signals and those of their peers, and how much such interference may actually negatively effect the groups' ability to navigate effectively. The authors use generalized linear models to test the significance of the effects they observe.The work relies on a thoughtful and detailed model which faithfully incorporates salient features, such as acoustic elements like the filter for a biological receiver and temporal aggregation as a kind of memory in the system. At the same time, the authors abstract features that are complicating without being expected to give additional insights, as can be seen in the choice of a two-dimensional rather than three-dimensional system. I thought that the level of abstraction in the model was perfect, enough to demonstrate their results without needless details. The results are compelling and interesting, and the authors do a great job discussing them in the context of the biological literature.With respect to the first version of the manuscript, the authors have remedied all my outstanding questions or concerns in the current version. The new supplementary figure 5 is especially helpful in understanding the geometry.
**Recommendations for the authors:**

**Reviewer #1 (Recommendations for the authors):**
Data Availability: This reviewer lauds the authors for switching from a private commercial folder requiring login to one that does not. At the cost of being overtly pedantic - the Github repository is not a long-term archival resource. The ideal solution is to upload the code in an academic repository (Zenodo, OSF, etc.) to periodically create a 'static snapshot' of code for archival, while also hosting a 'live' version on Github.

We have uploaded to Zenodo repository, and updated the link in the paper:

How bats exit a crowded colony when relying on echolocation only - a modeling approach

In one of the rebuttals to Reviewer #3- the authors have cited a wrong paper (Beleyur & Goerlitz 2019) - while discussing broad bandwidth calls improving detection - and may wish to correct this if possible on record.

We have removed the incorrect citation from the revised version of the manuscript.

Specific comments on the 2nd manuscript:Figure 5: Table 1 says 1, 2,5,10,20,40,100 bats were simulated (line 138-139) but the conclusion (line 398) says '1 to 100 bats' per 3msq. However, the X-axis only stops at 40 and says 'number of bats', while the legend says bats/3msq....what is actually being plotted? Moreover, in the entire paper there is a constant back-and-forth between density and # of bats - perhaps it is explained beforehand, but it is a bit unsettling - and more can be done to clarify these two conventions.

While most parameters were tested across the full range of 1 to 100 bats per 3 m², a subset of conditions—including misidentification, multi-call clustering, wall target strength, and conspecific target strength—were simulated only up to 40 bats due to significantly longer run-times. This is now clarified in both the main text and the Table 1 caption.

In our simulations, the primary parameter was the number of bats placed within a 3 m² starting area, which directly determined the initial density (bats per 3 m²). Throughout the manuscript, we use “number of bats” to refer to the simulation input, while “density” denotes the equivalent ecological measure. Figure 5 and related captions have been revised accordingly to note these conventions and to indicate when results are shown only up to 40 bats (see lines 120–122, 314-317 in the revised text).

Table 1: This was made considerably difficult to read given the visual clutter - and I hope I've understood these changes correctly.What is in the square brackets of the effect-size e.g. first row with values 'Exit prob. (%)' says -0.37/bat [63:100] ? What does this 63:100 refer to?What is the 'process flag'

Values in square brackets indicate the minimum and maximum values of the metric across the tested range (e.g., [63:100] shows the range of exit probabilities observed across different bat densities).

The term “process flag” has been replaced with “with and without multi-call clustering” for clarity

Both the table layout and caption have been revised to reduce visual clutter and to make these conventions clearer to the reader.

Lines 562-3: "In our study, due to the dense cave environment, the bats are found to operate in the approach phase nearly all of the time, which is consistent with natural cave emergence behavior" - bats are 'found to' implies there is some experimental data or it is an emergent property. See above for the point questioing the implementation of multiple echolocation phases in the model, but also - here the bat-agents are allowed to show different phases and thus they do so -- it is a constraint of the implementation and not a result per se given the size of the cave and the number of bats involved...

We removed the sentence from the Methods section, since it could be misinterpreted as an experimental finding rather than a model outcome. Instead, we now discuss this in the Discussion, clarifying that the predominance of the approach phase arises from the cluttered cave environment in our simulations, which is consistent with natural emergence behavior (see lines 355-363). In this context, the use of echolocation phases is presented as a biologically plausible modeling choice rather than an empirical result.

Lines 659-660: The parametrisation between DoA and SNR is supposedly found in 'Equation 10' - which this reviewer could not find in the manuscript

The equation was accidentally omitted in the previous revision and has now been reinserted into the manuscript. It defines how direction-of-arrival (DoA) error depends on SNR and azimuth angle (see lines 603-605).